# Parental Self-Efficacy as a Predictor of Children’s Nutrition and the Potential Mediator Effect between the Health Promotion Program “Join the Healthy Boat” and Children’s Nutrition

**DOI:** 10.3390/ijerph17249463

**Published:** 2020-12-17

**Authors:** Ricarda Möhler, Olivia Wartha, Jürgen Michael Steinacker, Bertram Szagun, Susanne Kobel

**Affiliations:** 1Faculty Social Work, Health & Nursing, University of Applied Sciences Ravensburg-Weingarten, 88250 Weingarten, Germany; ricardamoehler@arcor.de (R.M.); szagun@rwu.de (B.S.); 2Division of Sports and Rehabilitation Medicine, Department of Internal Medicine, University Hospital Ulm, 89075 Ulm, Germany; olivia.wartha@uni-ulm.de (O.W.); juergen.steinacker@uniklinik-ulm.de (J.M.S.)

**Keywords:** parental self-efficacy, children, diet, nutrition, health promotion, overweight, obesity

## Abstract

Overweight and obesity, as well as their associated risk factors for diseases, are already prevalent in childhood and, therefore, promoting healthy eating is important. Parental self-efficacy (PSE) and early health-promotion can be helpful in promoting healthy eating. The aim of this study was to examine the influence of PSE on children’s nutrition behavior and identify PSE as a mediator between an intervention and children’s nutrition. The kindergarten-based health-promotion program “Join the Healthy Boat” was evaluated in a randomized controlled trial with 558 children (4.7 ± 0.6 years; 52.3% male) participating at both times. Linear and logistic regressions as well as mediation analyses with potential covariates such as parental outcome expectancies or parental nutrition were carried out using questionnaire data. In children, PSE was positively associated with fruit and vegetable intake (*β* = 0.237; *p* < 0.001) and showed a protective effect on soft drink consumption (*OR* 0.728; *p* = 0.002). Parental nutrition was a stronger predictor of children’s intake of fruit, vegetables (*β* = 0.451; *p* < 0.001), and soft drinks (*OR* 7.188; *p* < 0.001). There was no mediator effect of PSE. However, outcome expectancies were associated with PSE (*β* = 0.169; *p* = 0.032). In conclusion, interventions should promote self-efficacy, outcome expectancies, and healthy nutrition for parents as well in order to strengthen the healthy eating habits of children.

## 1. Introduction

Non-communicable diseases such as cardiovascular diseases, cancer, or diabetes are one of the main causes of global deaths [1]. The risk factors of these diseases are overweight and obesity as well as hypertension, lack of physical activity, and an unhealthy diet, which all contribute to one other [2].

In 2017, an unhealthy diet was the cause of 11 million deaths and 255 million disability-adjusted-life-years [3]. Increasing the consumption of fruit and vegetables could help prevent overweight [4], diabetes [5], cancer [6], and cardiovascular diseases [7]. Conversely, the reduction of sugar sweetened beverages is known to reduce the risk of the above mentioned diseases [8], as well as the risk of fatty liver disease [9].

A healthy diet is vital for children because, on one hand, it contributes to a healthy cognitive development [10,11,12] or even a higher IQ and memory [13]. On the other hand, it can prevent risk factors for overweight and obesity in children and already prevalent consequences such as problems with puberty [14], diabetes, metabolic diseases, fatty liver disease, or coronary heart disease [14,15]. Psychological consequences, lower levels of education, and stigmatization are also associated with childhood overweight [14,16].

There seems to be a stagnation or even a decline of childhood obesity in developing countries, which public health programs have contributed to [15,17]. However, a recent U.S. study showed that the prevalence of obesity in U.S. children at any age [18] and extreme obesity in general are still increasing [17,18]. These findings highlight the importance for prevention and health promotion in early childhood. Starting health promotion as early as possible is essential because eating habits learned during childhood have been shown to continue into adolescence and adulthood [10,19]. The period from birth to preschool is considered very important because most nutrition habits are established then [19,20].

Even though genetic predispositions play a role in the development of risk factors, family and (social) environment can affect children’s eating habits [21]. Parents affect their children in their preferences and intake with the food they provide [22,23], their own intake [23,24], and joint family dinners [25]. Parenting habits such as pressure, restrictions, or reward hinder the development of healthy eating behaviors; instead, being a positive role model and promoting children in self-regulation have been associated to the development of healthy eating habits [21].

It has been suggested that parental self-efficacy is a predictor for children’s adjustment [26], and it therefore potentially influences children’s dietary habits. Self-efficacy can be defined as “people’s beliefs about their capabilities to produce designated levels of performance that exercise influence over events that affect their lives” [27]. Empirically, it was found that the expectation of self-efficacy plays a significant role in the initiation, maintenance, or control of health behaviors as well as the avoidance of health risks [28,29,30,31]. In exercise, diet, weight control, smoking, alcohol consumption, or drug use, as well as in pain therapy or stress management, the expectation of self-efficacy is a decisive predictor [29,30,31]. Recent research shows that, for diabetes patients, self-efficacy and outcome expectancies can be a good predictor of nutritional behavior [28].

On the other hand, parental self-efficacy has been shown to be helpful for a positive parent-child relationship, parental competence, educational quality, and (mental) parental health, as well as child development and adaptability [26,32,33]. A positive influence of parental self-efficacy on children’s health behavior, such as dental hygiene [34], exercise behavior [35,36], and the consumption of healthier food such as fruit and vegetables, and drinking, as well as the avoidance of unhealthy food has also been found [35,36,37,38]. 

Beside parental influence, kindergarten has been shown to be an ideal setting for health promotion because most children in Germany visit kindergarten during the period from birth to school entry [39]. There, a collaboration between family and other institutions in the child’s life is easily realized, and there is a higher range of possibilities in implementing interventions than at school because of a less strict curriculum [40].

“Join the Healthy Boat” is a health promotion program for kindergarten children. Implemented in 2014 in southwest Germany, it aims to promote healthy eating, increased physical activity, and meaningful leisure time activities using behavioral and structural prevention aspects. The program was developed using the intervention mapping approach [41] and is based on Bandura’s social cognitive theory [42] and Bronfenbrenner’s socio-ecological approach [43]. Therefore, it includes the promotion of children’s self-efficacy for health directed behavior change [44]. Because parents are informed through the kindergarten teachers and indirectly through children’s narratives from kindergarten, parental self-efficacy could also be affected.

In order to gain more theoretical knowledge about the potential of the construct of parental self-efficacy as well as predictors of children’s nutrition behavior, the aims of this study were to examine whether parental self-efficacy affects children’s nutrition, such as fruit and vegetable intake, as well as their consumption of sugar sweetened beverages.

To date, there is limited research available on parental self-efficacy and early health-promotion in the kindergarten setting. The health promotion program “Join the Healthy Boat” offers an intervention that intends to positively influence children’s nutrition behavior, but by including the family environment, the intervention potentially affects parents’ self-efficacy as well. In addition, this study tried to identify parental self-efficacy as a potential mediator between the intervention “Join the Healthy Boat” and children’s nutrition.

## 2. Methods

### 2.1. Intervention

The kindergarten intervention “Join the Healthy Boat” was developed using the Intervention-Mapping-Approach [41]. Based on a need assessment, concrete program targets were specified. Using theoretical models, such as Bandura’s socio-cognitive theory [42] and Bronfenbrenner’s socio-ecological approach [43], applications for the program were established and, based on these steps, the intervention program was conceptualized. Lastly, the implementation and evaluation of the program were planned based on the above mentioned framework [44]. 

Similar to the associated primary school intervention [45], the key topics of the program are increasing physical activity and meaningful leisure time, including a reduction of screen time as well as establishing a healthy diet, which is targeted by increasing fruit and vegetable intake as well as reducing the consumption of sugar sweetened beverages. The ready prepared teaching materials, which consist of 20 exercises and games lessons as well as 30 ready-to-use-ideas, were conceptualized for weekly use in kindergartens. Through healthy eating, physical activity, and leisure time units, children can explore alternatives for their behavior, gain knowledge about their bodies and health as well as improving motor skills, physical activity, and eating habits. Additionally, 5–7-min activity breaks, which can be used twice a day, are implemented. In order to gain an accepted and sustainable implementation of the program, kindergarten teachers were trained by a train-the-trainer-approach, receiving the materials, including instructional behavioral and educational resources, for free. To include all parents, parents’ evenings took place and parents’ letters were provided in three different languages [46].

Bespoke materials for parental work were designed appealingly, were drafted in easy and plain language, and offered specific action alternatives as well as health-related information. Kindergarten staff were provided with ideas and opportunities (posters designed by children, activities such as a joint healthy breakfast, parental letters, etc.) to present health relevant information to parents without them having to feel imposed upon.

### 2.2. Study Design

Data used for this study was retrieved from the evaluation study of the kindergarten program “Join the Healthy Boat”, which is a prospective, stratified, cluster-randomized longitudinal study with a control and an intervention group. Sixty-two kindergartens were randomized equally into a control or an intervention group. After baseline measurements, the intervention was implemented into the daily life of the kindergartens in the intervention group for one year. Follow-up measurements were conducted after that year, after which the kindergartens in the control group also implemented the program. More details of the evaluation study on recruiting, stratification, etc., are described elsewhere [46].

### 2.3. Ethics Approval and Consent to Participate

The evaluation study was registered in the German Register of Clinical Trials (DRKS-ID: DRKS00010089). Ethics approval was obtained from the university’s ethics committee and the Ministry of Culture and Education and was carried out according to the Declaration of Helsinki. Parents gave their written informed consent for the study, and the children gave their oral assent on site [46].

### 2.4. Study Population

At baseline, 973 children in 57 kindergartens aged 3 to 5 years (M = 3.6; SD = 0.6) participated. The final distribution of the intervention and the control group was 30 to 27 kindergartens, respectively, due to the drop-out of 5 kindergartens. At follow-up, a total of 558 (57%) children aged 4 to 6 years (M = 4.7, SD = 0.6) participated and were used for data analysis. More than half of the children were male (52.3%), 1/3 had a migration background (33.4%), and almost half of the children came from families with a high education level (48.7%). 

### 2.5. Instruments

The following instruments and methods were used for the data analysis of this study; other instruments of the intervention study are described elsewhere [47]. Using a parental questionnaire, sociodemographic data and dietary intake of parents and children were assessed. The items were taken from the German Health Interview and Examination Survey for Children and Adolescents (KiGGS) [48]. In addition, parental self-efficacy and parental outcome expectancies on their children’s dietary intake were questioned using other validated instruments [49], translated into German.

Migration background was operationalized as a dichotomous variable with one level as “Children with at least one parent born in a foreign country or children who were spoken to in a foreign language for the first three years or their life” based on the definitions of the Federal Statistical Office [50] as well as Settelmeyer and Erbe [51]. As an indicator for socioeconomic status, family education level was operationalized using the CASMIN-Classification [52], where school education and university/college degrees were weighted with each other. Tertiary education level was the highest, and primary and secondary education levels were grouped into one category. 

To include the potential specificity of self-efficacy beliefs [53], parental self-efficacy was not operationalized into one large scale, but into two scales to distinguish between parental self-efficacy on children’s fruit and vegetable intake as well as parental self-efficacy on children’s sugar sweetened beverage consumption. To gain a better comparability between parental self-efficacy and parental outcome expectancies, outcome expectancies were distinguished into two scales as well.

The scales regarding the effects of parental self-efficacy on children’s intake of fruit and vegetables as well as on children’s consumption of sugar sweetened beverages were assessed with six items each [49] and were conceptualized as, for example, “I manage to ensure that my child eats fruit and vegetables daily, even if my child rather wants to eat sweets or fast food” with a seven-point Likert scale.

The scales for parental outcome expectancies on children’s intake of fruit and vegetables as well as sugar sweetened beverages were assessed with six items each and conceptualized as, for example, “If my child drinks less sugary sweetened beverages, it is good for their teeth” with a seven-point Likert scale. More detailed information on the scales are shown in Appendix A.

The dietary intake of children, mothers, and fathers were assessed with four items each, querying the portions of fruit, raw and cooked vegetables, and vegetable juice consumed during the week (“never” to “three times per day or more”). The intake of sugar sweetened beverages was assessed by one item each, asking if sugar sweetened beverages were consumed “never” to “repeatedly during the week”. To have a better comparison, mothers’ and fathers’ dietary information was combined and calculated as parental intake. Fruit and vegetable intake was calculated as a metric variable whereas consumption of sugar sweetened beverages was calculated as a dichotomous variable, with one level as “sugar sweetened beverages once a week or more”. 

### 2.6. Data Analysis 

Multiple linear regression analyses for the metric dependent variable “children’s fruit and vegetable intake” were carried out. Logistic regressions were used to analyze associations between parental self-efficacy and children’s intake of sugar sweetened beverages at least once a week or more. Several requirements for regression analyses [54,55,56] had been examined and tested previously. The models were chosen based on theoretical deliberations. Parental outcome expectancies as well as parental nutrition were used as covariates to specify the predictors of children’s nutrition behavior [23,24,31]. Age, gender, migration background, and socioeconomic status can influence children’s nutrition behavior [57], which is why these were included as covariates as well. For a better comparability between the two dependent variables, the same control variables were used for both analyses.

Two different analyses tested whether parental self-efficacy is a predictor of children’s intake of fruit and vegetables or sugar sweetened beverages, respectively. It was controlled for age, gender, migration background, family education level, parental intake of fruit and vegetables, and parental consumption of sugar sweetened beverages, as well as outcome expectancies on children’s intake of fruit and vegetables and sugar sweetened beverages.

To examine a potential mediator effect of parental self-efficacy, the interaction effect between intervention, parental self-efficacy, and children’s intake of fruit and vegetables as well as sugar sweetened beverages was tested for significance. The mediation analyses were carried out with PROCESS [58]. PROCESS is a macro for SPSS (SPSS Inc., Chicago, IL, USA) that uses linear regression analyses for metric dependent variables or logistic regression analyses for dichotomous dependent variables to determine the direct and indirect effects for mediator and moderator models or the combination of both [58]. 

Three different regression analyzes for each dependent variable (children’s fruit and vegetable intake, children’s intake of sugar sweetened beverages) were carried out using the control variables of age, gender, family education level, migration background, and parental outcome expectancies. To identify the potential effect of the intervention on parents or children, parental self-efficacy and children’s nutrition values at baseline were used as control variables as well. One analysis examined the effect of the intervention on parental self-efficacy, the other two examined the role of the intervention on children’s nutrition with and without the mediator as a control variable.

Based on these calculations, a significance test for the indirect effect using the bootstrapping method and its generated confidence intervals was performed. Preacher and Hayes [59] recommend generating at least 5000 samples for the bootstrapping method, which is why 10,000 samples were chosen for this mediation analysis. Before the mediation analysis, several requirements for regression analysis [54,55,56] were examined and tested. Heteroskedasticity-consistent standard error estimators were used, because the preliminary testing did not exclude heteroskedasticity [60].

## 3. Results

Children’s and parents’ characteristics are presented in Table 1. At baseline, there were no significant differences between the intervention and the control group for the reported variables, apart from gender and migration background, with significantly more boys and more children with a migration background in the control group (*p* = 0.031 and *p* = 0.028, respectively).

### 3.1. Parental Self-Efficacy as a Predictor of Children’s Nutrition

#### 3.1.1. Parental Self-Efficacy as a Predictor of Children’s Fruit and Vegetable Intake

Parental self-efficacy on children’s fruit and vegetable intake (*ß* = 0.237 [0.083; 0.190] *p* < 0.001) as well as parental intake of fruit and vegetables (*ß* = 0.451 [0.391; 0.597] *p* < 0.001) were positive predictors of children’s intake of fruit and vegetables, even though parental consumption has a slightly higher standardized regression coefficient. No other variable had a significant effect on children’s fruit and vegetable intake (see Table 2).

#### 3.1.2. Parental Self-Efficacy as a Predictor of Children’s Consumption of Sugar Sweetened Beverages

Parental self-efficacy on children’s intake of sugar sweetened beverages (odds ratio of regression (*OR*) 0.728 [0.594; 0.891] *p* = 0.002) as well as tertiary family education level (*OR* 0.583 [0.345; 0.984] *p* = 0.043), and being female (*OR* 0.469 [0.277; 0.793] *p* = 0.005) had an inverse effect on children’s intake of sugar sweetened beverages. On the other hand, if at least one parent consumed sugar sweetened beverages at least once a week or more, the chance that children consumed sugar sweetened beverages at least once a week or more increased significantly (*OR* 7.188 [95% CI: 3.896; 13.264] *p* < 0.001; see Table 3).

### 3.2. Parental Self-Efficacy as a Mediator between the Intervention “Join the Healthy Boat” and Children’s Nutrition

#### 3.2.1. Parental Self-Efficacy as a Mediator between the Intervention “Join the Healthy Boat” and Children’s Fruit and Vegetable Intake

Individual analyses showed no significant effect of parental self-efficacy (*b* = 0.043 [−0.013; 0.099] *p* = 0.13) on children’s intake of fruit and vegetables. There was no effect of the intervention on parental self-efficacy (*b* = −0.185 [−0.434; 0.064] *p* = 1.145) or children’s intake of fruit and vegetables (*b* = −0.032 [−0.156; 0.091] *p* = 0.609). The non-standardized indirect effect between the intervention, parental self-efficacy, and children’s fruit and vegetable intake was −0.008 with a 95%-confidence interval of −0.029 to 0.003, which shows no significant effect either. 

However, there was a significant effect of some control variables. Parental outcome expectancies had an effect on parental self-efficacy on children’s intake of fruit and vegetables (*b* = 0.169 [0.014; 0.323] *p* = 0.032), and parental intake of fruit and vegetables had an effect on children’s intake of fruit and vegetables (*b* = 0.306 [0.208; 0.404] *p* < 0.001). Some baseline control variables affected variables at follow-up (see Figure 1).

#### 3.2.2. Parental Self-Efficacy as a Mediator between the Intervention “Join the Healthy Boat” and Children’s Fruit and Vegetable Intake

In the individual analyses, a significant effect of parental self-efficacy on children’s intake of sugar sweetened beverages could be observed (*b* = −0.296 [0.560; 0.032] *p* = 0.028). There was no significant effect of the intervention on the mediator parental self-efficacy (*b* = 0.069 [−0319; 0.183) *p* = 0.592) nor on children’s intake of sugar sweetened beverages (*b* = 0.067 [−0.521; 0.656] *p* = 0.823). The non-standardized indirect effect between the intervention, parental self-efficacy, and children’s intake of sugar sweetened beverages was −0.020 with a 95%-confidence interval of −0.064 to 0.123, which showed no significance either. 

However, parental consumption of sugar sweetened beverages was one of the highest predictors for children’s intake of sugar sweetened beverages (*b* = 1.602 [0.939; 2.266] *p* < 0.001). There was no significant effect of gender or family education level, compared to the first analyses. Some baseline control variables affected variables at follow-up (see Figure 2). 

## 4. Discussion

The first aim of this study was to show whether parental self-efficacy affects the nutrition behavior of kindergarten children. By including several covariates into the analyses, the predictors of children’s nutrition behavior could be specified. The second aim of the study was to show whether parental self-efficacy was a mediator between the intervention “Join the Healthy Boat” and children’s nutrition behavior. 

The results of this study show that parental self-efficacy is a predictor of children’s nutrition behavior, although parental nutrition seems to be a higher predictor for children’s nutrition behavior. Additionally, parental self-efficacy was not a mediator between intervention and children’s nutrition behavior. However, results of the mediation analyses showed that parental outcome expectancies on children’s fruit and vegetable intake were a predictor for the effects of parental self-efficacy on children’s fruit and vegetable intake. 

The findings of this study show that parental self-efficacy had a positive effect on the fruit and vegetable intake of their children, which can partially be confirmed by results of other studies [35,38]. Regarding children’s consumption of sugar sweetened beverages, an Australian study [38] showed that maternal self-efficacy influenced children’s water intake, whereas a Swedish study [35] reported no significant effect of maternal self-efficacy on children’s intake of sugar sweetened beverages. One explanation for the significant effect of parental self-efficacy on children’s intake of sugary beverages could be that parental self-efficacy was operationalized differently. The here analyzed study dealt with parental self-efficacy on fruit and vegetable consumption and consumption of sugar sweetened beverages separately whereas, in other studies [35,38], it was summarized together as parental self-efficacy.

In the mediation analysis, parental self-efficacy on fruit and vegetable intake lost its effect after adding the variable intervention and baseline values of parental self-efficacy as well as children’s fruit and vegetable intake. It is known that baseline values of health behaviors can affect follow-up values, which might be the reason for this. Because controlling for baseline values was irrelevant for the first hypothesis (“parental self-efficacy is a predictor for children’s fruit and vegetable intake”) the effect of parental self-efficacy on children’s fruit and vegetable intake can be confirmed with reservation.

Family education level and children’s gender showed significant associations with the consumption of sugar sweetened beverages, but not with their fruit and vegetable intake. Similar findings were observed in the KiGGS-study, which found significant gender differences in fruit and vegetable intake in 11- to 17-year-olds, but none in 3- to 10-year-old children [61]. In contrast, there are significant differences between boys and girls in all age groups (3- to 17-year-olds) regarding the consumption of sugar sweetened beverages in favor of girls showing a lower consumption of sugar sweetened beverages [61]. 

Moreover, it has been shown that children from families with low socioeconomic status are more likely to consume less fruit and vegetables and to drink more lemonade [10]. This could not be demonstrated here because family education level, as an indicator of socioeconomic status, only has a significant impact on the consumption of sugar sweetened beverages and not on children’s fruit and vegetable intake. It has to be noted that, in other studies, socioeconomic status was assessed based on the level of family education, professional position, and income [62,63]. On the other hand, other studies mostly consider low, medium, and high socioeconomic status [62,63] whereas, in this work, low and medium family education levels were grouped into one category to be compared to high family education level. As a result, secondary family education level could have cancelled out any significant influence of those with primary family education level. Ultimately, it should be noted that differences of gender and socioeconomic status were lost in the mediator analysis. However, this can be explained by the significant effect of the baseline values of children’s intake of sugary beverages on the follow-up values of children’s intake of sugary beverages (Figure 2), which are irrelevant for the first hypothesis (“parental self-efficacy is a predictor for children’s nutrition behavior”). Therefore, the effects of gender and socioeconomic status differences on consumption of sugar sweetened beverages should be viewed with caution.

Age, migration background, or parental outcome expectancies did not affect children’s nutrition at follow-up. In an Australian study [38], the level of parental self-efficacy as well as children’s nutrition was linked to age. One- and 5-year-old children were compared in a cross-sectional design, so the difference between those children could have arisen due to the study design and as well as the large age difference. Other studies with significant age differences on nutrition compare older children or a wider range of age groups, which could explain the missing differences between 4- to 6-year-olds in this study [61,64]. 

It is known that migration background can impact on children’s nutritional behavior [10]. However, the difference depends on the country of origin [65,66] and how long the families have been living in Germany [10]. Therefore, it is possible that children examined in this study come from families who have not yet lived in Germany for a long time or come from countries of origin who do not have such unfavorable eating patterns. In addition, migration background was operationalized as “Children with at least one parent born in a foreign country or children who were spoken to in a foreign language for the first three years of their life”. In other studies, this is defined as one-sided migration [67]. In analyses in which differences between children with and without migration backgrounds were found, two-sided migration background—i.e., if both parents or the child and one parent was born abroad—was mostly examined [65,66].

In diabetes patients, outcome expectancies combined with the self-efficacy of those patients were the strongest predictors of their own nutritional behavior [28]. Therefore, the hypothesis was that parental self-efficacy, combined with parental outcome expectancies, would have an influence on children’s behaviors. However, in this study, parental outcome expectancies showed no relationship with children’s intake of fruit, vegetable, and sugar sweetened beverages. Because this study relates to the effects of parental self-efficacy on children’s nutrition behavior, it may be that outcome expectancies have no direct impact on children’s behavior but do influence parents’ own behaviors. This can be explained with the theoretical consideration that outcome expectancies are a preliminary stage of self-efficacy and therefore play a role in the intention formation for one’s own behavior change [68]. This is evidenced by the significant influence of parental outcome expectancies of children’s fruit and vegetable intake on the effects of parental self-efficacy on children’s fruit and vegetable intake in the mediation analyses. Even though this study could not support the hypothesis that parental outcome expectancies affect children’s nutrition behaviors, the result is important because Williams [69] points out a contradiction in Bandura’s theory of self-efficacy [53]. Bandura [70] says that outcome expectancies have no causal influence on self-efficacy, but self-efficacy remains valid even if they were influenced by outcome expectancies. However, because experimental studies have shown that outcome expectancies have an impact on self-efficacy [71,72], they should be considered as well, and a distinction should be made in the theory of self-efficacy as to whether outcome expectancies have an impact on self-efficacy [69]. The results of this work could make a further contribution to that.

It should be noted that parental intake of fruit, vegetables, and sugar sweetened beverages has a significant to a highly significant effect across all analyses and, in contrast to other influencing variables, has the highest contribution to children’s intake of fruit, vegetables, and sugar sweetened beverages. Parents’ own consumption can be seen as an example of what to eat, but will also reflect what is available at home. Both components are key influences for children’s nutrition [22,23,24]. So far, other studies that examined parental self-efficacy as a predictor of children’s eating behavior have not included parental nutrition [38] or have concluded that maternal nutrition does not affect the relationship between self-efficacy and children’s nutrition [35]. However, the statistical models that included maternal nutrition as well were the ones with the highest explained variance [35]. Because bespoke studies only examined maternal consumption, the different results of this work can be explained by the summarized values of mothers and fathers regarding parental intake of fruit and vegetables as well as sugar sweetened beverages. 

Parental self-efficacy could not be identified as a mediator between the health promoting intervention and children’s nutrition, which might be explained by the disappearance of the significant influence on fruit and vegetable intake, as well as by the fact that the intervention had neither a significant effect on the mediator nor on the children. In the evaluation study of the program “Join the Healthy Boat” [47], no significant differences between the intervention and the control group were found with regards to consuming fruit, vegetables, and sugar sweetened beverages. However, the percentage of sugar sweetened beverages consumed daily or several times a day decreased by 4.8% in the intervention group at follow-up, whereas, in the control group, it remained the same [47]. In this work, intake of sugar sweetened beverages was divided into two categories: “less than once a week” and “once a week or more”. Because, in this study, the intervention has a marginal but positive coefficient on the consumption of sugar sweetened beverages once a week or more, this could indicate that the intervention may only have a potentially reducing effect when consuming sugar sweetened beverages more often. Ultimately, other control variables were used in the evaluation study, which could be an explanation for the inconsistencies within this study [47]. 

The intervention had no significant effect on parental self-efficacy, which could be due to parental involvement only through letters and parents’ nights; there was no so-called family homework as implemented in the intervention for primary school children [73]; therefore, parental or family inclusion might have been insufficient for a significant influence on parental self-efficacy. Nevertheless, the results indicate a potential and a need for parents being included in interventions because of their significant influence on kindergarten-aged children through parental self-efficacy and parental consumption of fruit, vegetables, and sugary beverages.

In order to correctly interpret the results discussed, the strengths and limitations of this work should be considered. One strength of this study lies in the underlying data from the evaluation study. With its cluster-randomized waiting control group design, valid statements can be made about causal relationships in relation to the predictors of children’s health behaviors. The high and diverse sample, as well as the recruitment across the southwest of Germany, strengthens the study. However, the southwest is rather well-off compared to other states, which is also reflected in the high proportion of families with tertiary education levels in the sample, so that a generalization to other federal states is not necessarily possible. Furthermore, a selection bias cannot be ruled out, because mostly kindergartens with an affinity for health promotion as well as health-conscious and well-educated parents agreed to participate in the study. In terms of method, it should be noted that the measures are based on self- and external assessments of the parents regarding their own and their children’s behaviors, which were not tested again for reliability and validity. A source of error could be social desirability as well as the wrong estimations of children’s nutrition while being at kindergarten.

The mediation analysis with PROCESS should be mentioned as a strength, because the problems with the practice postulated by Baron and Kenny [74] can be reduced [75,76,77,78]. On the other hand, it is a new method and the confidence intervals are generated using percentile bootstrapping and not error-corrected bootstrapping, which tries to map the correct centering of the confidence intervals more appropriately. The percentile bootstrapping method requires a larger sample than the error-corrected one in order to have a sufficient power of 0.8 [79]. However, Hayes and Scharkow [80] note that, even though error corrected bootstrapping should be used when power is the biggest problem, percentile bootstrapping is a good compromise between power and error of the first type. Therefore, percentile bootstrapping used for the investigation appeared to be appropriate for this sample size.

Another point that should be noted positively is the consideration of parental self-efficacy in relation to children’s individual behaviors because this enables differentiated statements about the effect and potential influence of the construct of parental self-efficacy. However, as mentioned at the beginning, parental self-efficacy poses a definition difficulty. Because it is derived from the general theory of self-efficacy without having its own theoretical framework [26], it is generally difficult to replicate and compare with other studies. In this work, the items for the assessment of parental self-efficacy and parental outcome expectancies were taken from a non-German instrument, which meant they had to be translated and were then not tested again for their validity and reliability, which represents a further limitation. In the mediation analysis, parental self-efficacy on children’s fruit and vegetable consumption lost its significant effect. Because no hierarchical regression was carried out, it cannot be said if this was due to the inclusion of baseline values or if it was for other reasons. As a further limitation of this study, it should be noted that fruit and vegetable intake was not considered separately, which might have led to misinterpretations because parental self-efficacy for promoting fruit intake behaviors may differ from those promoting vegetable intake behaviors.

The operationalization of individual variables should be changed for future research. For example, no further distinction was made between mothers and fathers. The level of family education was only considered in two categories and was used as the only indicator of socioeconomic status. Therefore, it would be important to differentiate the consumption, self-efficacy, and outcome expectancies between mothers and fathers because other studies showed special influences of the mothers [35,38]. For better comparability and more differentiated interpretation, the socioeconomic status should be recorded using other indicators as well, such as professional position and household income, as well as comparing the categories low, medium, and high. Because a German study [36] showed parental self-efficacy as a predictor of physical activity of primary school children, it would be important for future research to investigate whether this influence also exists in kindergarten children and if it is a mediator, because a positive effect of the kindergarten intervention “Join the Healthy Boat” was shown on physical activity [47]. Overall, no mediator effect of parental self-efficacy can be found, but because there was a significant effect of outcome expectancies on parental self-efficacy as well as the effects of parental intake of fruit, vegetables, and sugar sweetened beverages on children’s nutrition behavior, these potential predictors and possible mediators should be examined further.

## 5. Conclusions

In conclusion, in spite of the work’s limitations, parental self-efficacy can be seen as a predictor for children’s intake of fruit, vegetables, and sugar sweetened beverages. As a recommendation for the intervention ”Join the Healthy Boat” and other health promoting interventions, parents should be included more intensely. The focus should include parental nutrition, parental self-efficacy, and parental outcome expectancies, as this can have direct and indirect effects on children’s nutrition. Gender, age, migration background, and family education level had inconsistent or no effect, but this could be because differences appear later in life, which highlights the need for early prevention. In future research, parental self-efficacy on children’s physical activity in kindergarten children could be discussed to specify the potential effect and use it for the intervention “Join the Healthy Boat” and other health promoting interventions.

## Figures and Tables

**Figure 1 ijerph-17-09463-f001:**
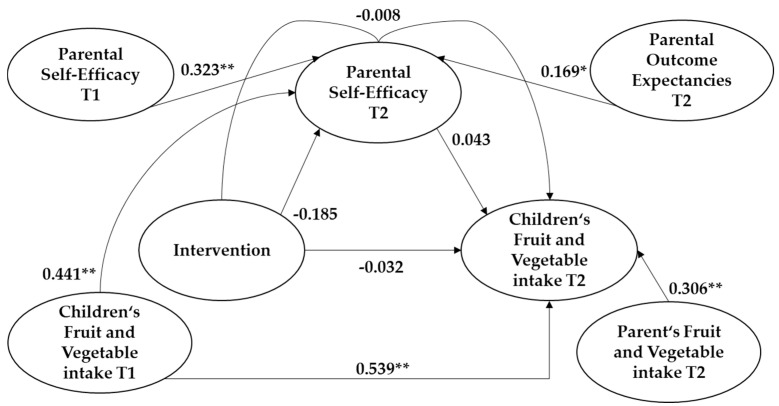
Results from the mediation analysis on children’s fruit and vegetable intake. (*) *p* < 0.05 (**) *p* < 0.001. T1 = baseline, T2 = follow-up.

**Figure 2 ijerph-17-09463-f002:**
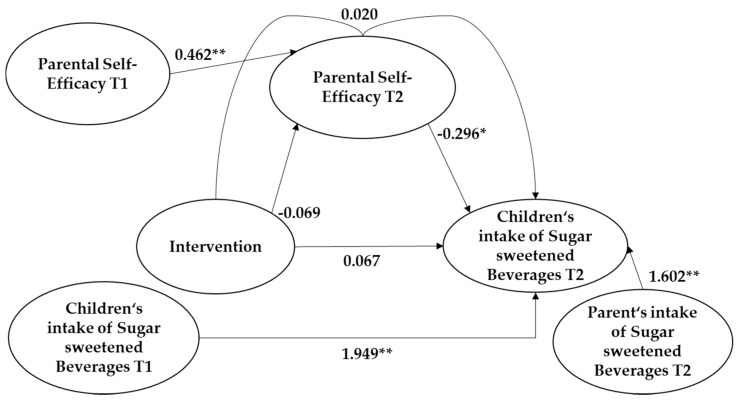
Results from the mediation analysis on children’s intake of sugar sweetened beverages. (*) *p* < 0.05 (**) *p* < 0.001. T1 = baseline, T2 = follow-up.

**Table 1 ijerph-17-09463-t001:** Children’s and parent’s characteristics divided into intervention and control group.

Variables	Missing Values	Intervention (*n* = 318)	Control (*n* = 240)	Total (*n* = 558)
Children				
Age at follow-up [*m*, *sd*]	3	4.7 (0.6)	4.6 (0.6)	4.7 (0.6)
Boys [*n*, %] ^1^		113 (47.1)	179 (56.3)	292 (52.3)
Migration background [*n*, %] ^1^	106	77 (30.6)	74 (37.0)	151 (33.4)
Fruit and vegetable intake * T1 [*m*, *sd*]	92	2.5 (0.9)	2.7 (0.7)	2.6 (0.8)
Fruit and vegetable intake * T2 [*m*, *sd*]	166	2.6 (0.8)	2.7 (0.8)	2.6 (0.8)
Sugar sweetened beverages once a week or more T1 [*n*, %]	94	101 (38.5)	77 (38.1)	178 (38.4)
Sugar sweetened beverages once a week or more T2 [*n*, %]	161	85 (38.6)	60 (33.9)	145 (36.5)
Parents				
Tertiary family education level [*n*, %]	106	120 (47.2)	100 (50.5)	220 (48.7)
Fruit and vegetable intake * T2 [*m*, *sd*]	168	2.4 (0.7)	2.5 (0.7)	2.4 (0.7)
At least one parent drinks sugar sweetened beverages once a week or more T2 [*n*, %]	162	127 (57.7)	106 (60.2)	233 (58.8)
Self-efficacy on children’s fruit and vegetable intake T1 [*m*, *sd*]	95	4.9 (1.6)	4.9 (1.4)	4.9 (1.5)
Self-efficacy on children’s fruit and vegetable intake T2 [*m*, *sd*]	163	4.8 (1.4)	5.0 (1.3)	4.9 (1.4)
Self-efficacy on children’s sugar sweetened beverage intake T1 [*m*, *sd*]	99	4.9 (1.5)	5.0 (1.4)	4.9 (1.4)
Self-efficacy on children’s sugar sweetened beverage intake T2 [*m*, *sd*]	168	4.9 (1.4)	5.1 (1.2)	5.0 (1.3)
Outcome expectancies on children’s sugar sweetened beverage intake T1 [*m*, *sd*]	162	6.3 (0.7)	6.2 (0.8)	6.3 (0.7)
Outcome expectancies on children’s fruit and vegetable intake T2 [*m*, *sd*]	161	6.0 (0.8)	5.8 (0.9)	5.9 (0.8)

Notes: T1 = baseline, T2 = follow-up, migration background = children with at least one parent born in a foreign country or children who were spoken to in a foreign language for the first three years or their life, * = mean of the scale (0 = never, 1 = less than once a week, 2 = 1–2 times a week, 3 = 4–5 times a week, 4 = once a day, 5 = 2 times a day, 7 = 3 times a day). ^1^ = significant difference between the control and the intervention group at baseline.

**Table 2 ijerph-17-09463-t002:** Regression analyses on parental self-efficacy and children’s fruit and vegetable intake.

Variables	*B* [95% CI]	*SE(B)*	*β*	*t*	*p*
Constant	0.744 [−0.126; 1.614]	0.442		1.682	0.094
Parents					
Self-efficacy	0.137 [0.083; 0.190]	0.027	0.237	5.005	<0.001
Fruit and vegetable intake	0.494 [0.391; 0.597]	0.052	0.451	9.466	<0.001
Outcome expectancies	0.015 [−0.071; 0.101]	0.044	0.016	0.341	0.733
Tertiary family education level	0.090 [−0.050; 0.230]	0.071	0.060	1.263	0.207
Children					
Age	−0.034 [−0.152; 0.084]	0.060	−0.027	−0.569	0.570
Gender	0.051 [−0.088; 0.190]	0.071	0.034	0.727	0.468
Migration background	0.004 [−0.148; 0.156]	0.077	0.002	0.053	0.958

Notes: Corrected R^2^ = 0.307 (*n* = 332, *p* < 0.001), *B* = non-standardized regression coefficient, CI = confidence interval of non-standardized regression coefficients, *SE(B)* = standard error of non-standardized regression coefficients, *β* = standardized regression coefficient, *t* = t-value of regression parameters, *p* = exact *p*-value tested on the 5%-level. Migration background = children with at least one parent born in a foreign country or children who were spoken to in a foreign language for the first three years or their life.

**Table 3 ijerph-17-09463-t003:** Regression analyses on parental self-efficacy and children’s intake of sugar sweetened beverages.

Variables	*B(SE)*	*p*	*OR* (95% CI)
Constant	0.458 (1.587)	0.773	1.581
Parents			
Self-efficacy	−0.318 (0.103)	0.002	0.728 (0.594; 0.891)
Intake of sugar sweetened beverages	1.972 (0.313)	<0.001	7.188 (3.896; 13.264)
Outcome expectancies	−0.110 (0.179)	0.537	0.896 (0.631; 1.271)
Tertiary family education level	−0.540 (0.267)	0.043	0.583 (0.345; 0.984)
Children			
Age	0.248 (0.224)	0.269	1.281 (0.826; 1.988)
Gender	−0.758 (0.268)	0.005	0.469 (0.277; 0.793)
Migration background	0.452 (0.285)	0.112	1.572 (0.900; 2.746)

Notes: R^2^ = 0.245 (Cox and Snell), 0.334 (Nagelkerke); Model χ^2^(7) = 93.58 (*n* = 333; *p* < 0.001), *B* = regression coefficient, *SE* = standard error of regression coefficients, *OR* = odds ratio of regression coefficients, CI = confidence interval of odds ratios, *p* = exact *p*-value tested on the 5% level. Migration background = children with at least one parent born in a foreign country or children who were spoken to in a foreign language for the first three years or their life.

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
