# Peer review of "Parental Self-Efficacy as a Predictor of Children’s Nutrition and the Potential Mediator Effect between the Health Promotion Program “Join the Healthy Boat” and Children’s Nutrition"

_ijerph, 2020, doi:10.3390/ijerph17249463_

Round 1
Reviewer 1 Report
Overall, the paper describes an interesting study, which could potentially make a contribution to the literature regarding parental influence on children’s eating behaviors. However, the primary concern is the lack of information about how parental self-efficacy and outcome expectancy scales were developed and tested, and what they actually measured. It was difficult to interpret the findings without a concrete idea of what these two constructs represented. Clarification of the methods is necessary. In addition, parts of the discussion section were confusing regarding whether the discussion was related to the baseline data analysis or the intervention outcomes, also which studies were being discussed, the current study or work by others.
Abstract
Lines 14-15: More accurate to say associated risk factors for diseases, rather than associated diseases (lines 40-41 indicate that children have problems with diabetes, metabolic diseases, fatty liver disease or coronary heart disease supported by a review [14] that indicates “Of these overweight children, a quarter are obese, with a significant likelihood of some having multiple risk factors for type 2 diabetes, heart disease and a variety of other co‐morbidities before or during early adulthood.” Lines 40-41 should also be revised accordingly.
Line 15-16: The previous statement is focused on promotion of healthy eating, therefore “in this regard” in lines 15-16 should be focused on healthy eating rather than the more general health promotion.
Lines 16-18: Include outcome expectancies in the aim.
Lines 21-22: Indicate why beta coefficients are used with FV intake and outcome expectancies and OR with soft drink intake
Line 26: Healthy nutrition for parents as well was not based on a result provided in the abstract.
Line 25 – were associated with rather than affect
Introduction
Line 39: Replace A healthy nutrition is… with A healthy diet is…
Lines 45-46: Reference 17 is based in part on older data from the US that is contradicted by the current increasing prevalence of overweight and obesity in the US. Need to revise.
Lines 57-61: Consider deleting this paragraph. The definition of self-efficacy and how it was conceptualized in this study follows lines 62-69.
Line 78: Clarify what is meant by interference between family and other institutions in the child’s life.
Line 83: Clarify what is meant by behavioral and structural preventing manner.
Methods
Lines 112-114: Expand on intervention components for parents to better describe the potential for influencing PSE.
Lines 141-142: Provide references for the other validated instruments.
Line 144: Revise: with whom was spoken to in a foreign language… Children with at least one parent from a foreign country or children who were spoken to in a foreign language…
Lines 155-159: Provide the items for all PSE and outcome expectancies scales in a separate table or supplementary table. Provide reliability and validity testing results for PSE and outcome expectancy scales. Using items from validated instruments does not guarantee reliability and validity when combined together in a new instrument. For the PSE scale on fruit and vegetable intake combined, note how this scale was developed given that PSE for fruit intake might be different than PSE for vegetable intake given that children are more resistant to eating vegetables than fruit. Lines 156-159 only provide the stem for the item and not the whole question, thus the reader does not know what the parent is rating their capability of doing even if some condition exists. It is difficult to interpret findings based on the baseline findings or the intervention findings when the reader doesn’t know what was being measured.
Line 164: Clarify how intake of mothers and fathers were calculated as parental intake (separately or combined?)
Lines 176-178: Describe how the covariates were selected for inclusion in the models for each analyses.
Table 1. Should indicate whether baseline characteristics were different between intervention and control groups. Looks like there may have been significant differences in several variables. Could list values for T1 before T2.
Discussion – some confusion throughout about whether the results being discussed were based on the first or second aim. Could better describe the context as points are discussed. Also could better clarify whether variables discussed are parent or child variables throughout (see lines 346-347 for example).
Lines 258-279 are a repetition of the results. Could summarize more succinctly, or instead repeat results in later paragraphs when they are specifically discussed.
Line 285: Not clear which study is being referred to as This study
Lines 294-306: This paragraph makes a good point about including parents’ consumption in models of PSE as a predictor of children’s intake, also that intake of both fathers and mothers needs to be considered in the analyses.
Lines 323: No information was provided about significance of baseline values. Please clarify.
Lines 351-358: Need to consider whether the studies cited were based on parent self-efficacy regarding promoting healthy behaviors for their children or based on an individual adult’s self-efficacy for their own behaviors. Would the discussion be the same?
Lines 373-376: An explanation of parent involvement in the methods section is necessary to clarify this statement. How was the intervention expected to influence PSE?
Lines 399-408: Need to discuss lack of testing for reliability and validity of PSE and outcome expectancies scales, if that is the case. Also in this study, fruit intake behavior was not considered separately from vegetable intake behavior, a limitation because fruit is generally better accepted by children than vegetables, and PSE for promoting fruit intake behaviors may not be the same as for promoting vegetable intake behaviors.
Reviewer 2 Report
Thank you for your interesting manuscript. I have some suggestions:
- introduction; could you indicate more clearly what is new in your research? Lines 72-75 already say that "a positive influence of parental self-efficacy on children’s health behaviour... was found". What was the reason for conduction another study on this topic?
- line 117: what stratification did you use and why?
- lines 137-142: please provide references for the questionnaires mentioned.
- lines 155-159: it would be helpful to see the questionnaire
- Table 1: not easy to read. sometimes T1 and T2 are reportes (e.g. last two items in table 1), sometimes only T1 or T2. Why? I would suggest to have the Table the other way around, so that is easy to compare differences between T1 and T2.
- What happened with missing values? For example: in children there are data for Sugar sweetened beverages once a week or more T2 N=283, compared to N=94 at T1. What data did you analyze?
- 3.2: What is the effect of the intervention? I cannot see a clear answer to that question. It is a lot of text and numbers. I think your message would be better of with less text and clearer comparison between the intervention and control group.
- lin 258: I think this should she: we aimed to investigate whether parentel self-efficacy affects nutrition behaviour... rather than aiming to find an effect.
- discussion: it is had to read, to be honest. It is a lot of text. I would suggest to highlight key messages that answer your research question. Your results are often in line with other studies. What makes this study interesting; what new information does it provide?
Round 2
Reviewer 1 Report
The authors have addressed my concerns, however, typos and incorrect grammar issues remain. Extensive editing for the English language and style should be completed to enhance readability.
